# Current Insights into the Formulation and Delivery of Therapeutic and Cosmeceutical Agents for Aging Skin

**Ayça Altay Benetti [1,\*]**, **Tamara Tarbox [2]** and **Camillo Benetti [1]**

1   Department of Pharmacy, National University of Singapore, Singapore 117559, Singapore
2   College of Pharmacy, University of Texas at Austin, Austin, TX 78712, USA
\*   Correspondence: ayca.ben@nus.edu.sg

**Abstract:** "Successful aging" counters the traditional idea of aging as a disease and is increasingly equated with minimizing age signs on the skin, face, and body. From this stems the interest in preventative aesthetic dermatology that might help with the healthy aging of skin, help treat or prevent certain cutaneous disorders, such as skin cancer, and help delay skin aging by combining local and systemic methods of therapy, instrumental devices, and invasive procedures. This review will discuss the main mechanisms of skin aging and the potential mechanisms of action for commercial products already on the market, highlighting the issues related to the permeation of the skin from different classes of compounds, the site of action, and the techniques employed to overcome aging. The purpose is to give an overall perspective on the main challenges in formulation development, especially nanoparticle formulations, which aims to defeat or slow down skin aging, and to highlight new market segments, such as matrikines and matrikine-like peptides. In conclusion, by applying enabling technologies such as those delivery systems outlined here, existing agents can be repurposed or fine-tuned, and traditional but unproven treatments can be optimized for efficacious dosing and safety.

**Keywords:** skin aging; microneedle; matrikines; peptides; cosmetics; SLNs; nanoparticles

## 1. Introduction

Skin is the protective physical barrier that protects our body against harm from the hazardous effects of ultraviolet (UV) radiation and the infiltration of pathogens, as well as the dehydration process. Skin aging can be divided into two types of processes: intrinsic, or chronological aging, and extrinsic aging. Intrinsic aging starts with the gradual degeneration of dermal tissue, including areas protected from sunlight; thus, genetic factors affect the process. Although excessive exposure to the harmful effects of environmental factors causes extrinsic aging, the explicit evidence of aging skin can be ascribed to morphological changes at early stages in life, leading to thin rhytides and creases, cutaneous dyschromia, and skin laxity, for example [1]. Physical aging mostly depends on the intensity of environmental factors combined with the influence of genetic factors [2]. As a consequence of all the possible factors affecting the aging process and the complex structural and functional transformations within the skin [3,4], there could be significant changes not only in the appearance, but also in the skin's overall health status [5,6]. Thus, numerous cosmetic and therapeutic agents have been investigated in terms of their potential to alleviate the undesirable effects of skin aging, with many available either over-the-counter (OTC) or by prescription [3,7,8]. Different types of agents proposed to confer anti-aging benefits include small-molecule drugs, vitamins, minerals, antioxidants, peptides, polysaccharides, and cell culture- and plant-derived products [8–14]. Given the range of physicochemical properties associated with these potential therapeutics, formulation strategies must be tailored not only to the specific agent, but also to the site of action, and the chosen delivery system needs to be designed to ensure safety and effectiveness in counteracting the negative effects of aging [11,15–17].

In this review, the purpose is to give an overall perspective on the main challenges in the formulation development of potential agents aimed at defeating or slowing down skin aging. Due to the complex structural and functional transformations of the skin, delivery systems of therapeutic and cosmetic agents have been studied for decades. Nevertheless, the strategies used thus far comprise a variety of applications which require targeted delivery modes in order to achieve good penetration into the skin.

### 1.1. Physiological Change in Aging Skin

Skin aging is associated with changes such as pigmentation that begin gradually and naturally progress over the years [18]. However, the aging process also encompasses potentially fatal diseases as well as increased dryness and irritation, elasticity loss, wrinkles, slower wound healing, diminished sensory input, actinic keratosis, and neoplasms such as cell carcinoma and malignant melanoma [5,6,19–23]. Considerable age-related changes occur in both the dermis and epidermis, which are the most external layers in the skin [24,25]. Structural transformations within the dermal extracellular matrix (ECM) that lead to clinically relevant changes in appearance are primarily attributed to major component remodeling, particularly in dermal collagens, elastic fibers, and glycosaminoglycans [3,8]. In addition to ECM remodeling, variations in epidermal thickness [24]; lipid, ion, water content, and surface pH changes [26]; and fewer melanocytes and Langerhans cells all contribute to diminished physical, chemical, and immunological barrier properties [6,19]. On the other hand, younger skin has the most coherent structure, as well as a thicker dermoepidermal junction and subcutaneous fat layer. Over the decades, the epidermis thickness (especially in women on the face, upper chest, and surface of the hands and forearms) decreases by approximately 6.4% [23]. In spite of the decrease in the epidermis thickness, the stratum corneum (SC) turns into a thicker and more rigid structure over the years, with relatively greater and flatter corneocytes that lead to intercellular cohesion due to altered degeneration of the corneodesmosomes, which are created by desmosomes with aging [27]. The desmosomes (Figure 1) are formed by keratinocytes, which consist of side-by-side connections between cells that maintain adhesion and act as a junction for the actin and tubulin cytoskeleton to move into the stratum granulosum and spinous spinosum [27,28]. This differentiation of the SC contributes to wound healing, although with aging, the reduction in desmosomes is affected, as is the healing speed [27]. Since desmosomes favor cell aggregation and help the skin to maintain its protective feature, their gradual destruction by proteolytic enzymes results in accumulation of keratinocytes on the skin. When the keratinocytes remain without any junction, such as desmosomes, their accumulation increases with aging and severe skin diseases, including palmoplantar keratoderma (PKK) and inflammatory peeling skin or desquamation, may occur [28].

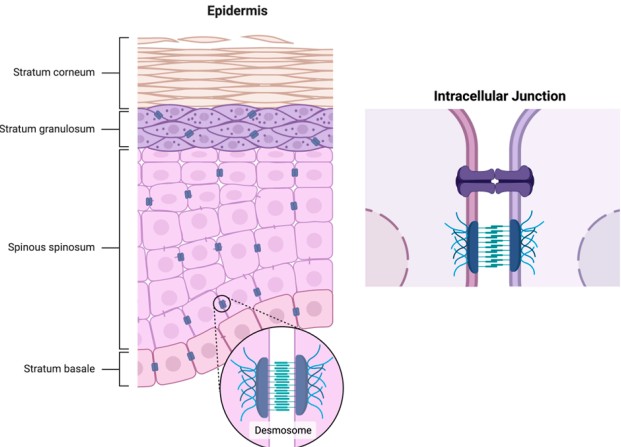

**Figure 1.** Desmosomes favor cell aggregation and connections (Reprinted from "desmosomal protein distribution in the epidermis", by BioRender [29]).

### 1.2. Primary Factors That Accelerate Aging

The primary environmental contributor to extrinsic skin aging is UV radiation from the sun (both long (UVA) and short (UVB) wavelength), or photoaging, though exposure to toxins from pollution and tobacco smoke contributes to it as well. Genetic factors have a greater influence on intrinsic individual aging, and with darker skin types there is less apparent photoaging than with fairer skin types due to the higher melanin content in the former [5,19]. Interestingly, according to the definition of Krutmann et al. [2], the environmental factors driving extrinsic skin aging also correlate with genetic factors, although the relevance of these factors seems to include different wavelengths other than UV radiation, as well as different air pollutants. On the other hand, the effects of aging have been compared to the formation and process of healing for wounds and laser procedures [14], with Hajem et al. observing that wound healing and aging have common molecular targets [30]. Altered cellular communication and replication, cellular senescence, hormonal changes, oxidative stress, and genetic mutations are all physiological factors that can promote age-related reparative effects in the skin [3,4,25,31–33]. The generation of reactive oxygen species (ROS) endogenously or induced by UV exposure has been identified as a primary contributor to aging, as has the upregulation of matrix metalloproteinases (MMPs), the enzymes responsible for degradation of the ECM. Mostly, the excessive amount of ROS enhances the upregulation of AP-1 (the activator protein) and NF-κB (nuclear factor), known as redox-sensitive transcription factors and mitogen-activated protein kinases (MAPKs) [5,25,31,32,34]. Once the mechanism of AP-1 is activated by low-dose UV exposure and further combines with the generation of ROS, the transforming growth factor β (TGF-β) receptors are suppressed and the synthesis of procollagen is blocked. Additionally, the collagen degradation caused by MMPs is triggered by the activation of AP-1 [34]. Although MMPs play an important role in the aging mechanism since their expression increases upon UV exposure, resulting in degradation of collagen at the level of the ECM, the mediator(s) that connect to the key MMPs associated with ECM degradation have not been thoroughly identified [35]. Some mediators related to this association, such as MMP-1 (collagenase), MMP-3 (stromelysin), and MMP-9 and MMP-2 (gelatinases), were studied in human skin. The results showed that MMP-9, MMP-3, and MMP-1 were activated upon UV exposure [5,36]. The study by Seok-Jin Lee et al. reported that transglutaminase 2 (TG2), which is known as one of five TG isozymes in the epidermal keratinocytes, plays a key role in the increase in MMP-1 expression when induced by UVB irradiation [35]. Another study focused on the determination of relevant mediators of photoaging and showed that the degradation of type IV collagen was followed by the intensive expression of MMP-2 in human skin. Additionally, while the induction of MMP-2 increases over the years in skin exposed to sunlight, it corresponds to the pathway of AhR (aryl hydrocarbon receptor). The same study also showed that the induction of MMP-2 and MMP-11 (another related photoaging mediator) was linked to the indication of the specificity protein 1 (SP1, which acts downstream of AhR) and MAPK pathways associated with DNA damage, whereas the mediators MMP-1 and MMP-3 were not correlated [37].

In addition to remodeling the dermal ECM, MMPs participate in various critical aspects of cell regulation, and thus, any treatments aimed at MMP inhibition must have a high target specificity to avoid undesirable effects [38]. Several MMPs have been implicated in fibrillin degradation within the ECM, but the cleavage of intact fibrillary collagen is uniquely initiated by MMP-1 [4,39,40]. A summary of the contributors to aging and their main consequences is illustrated in Figure 2.

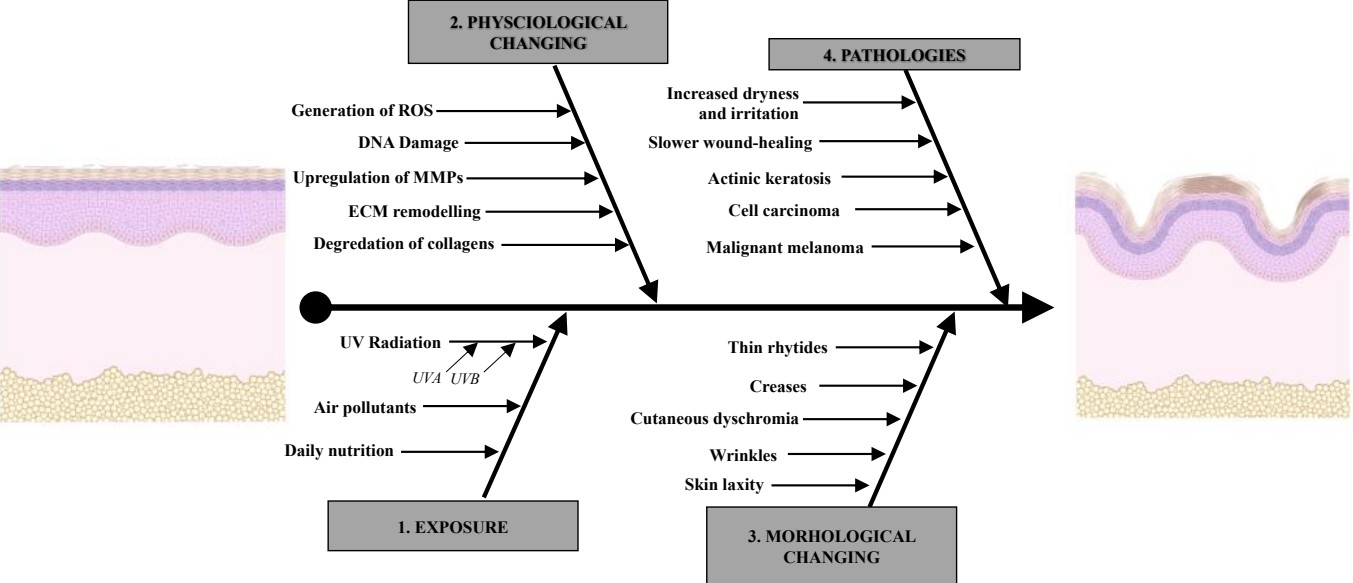

**Figure 2.** Aging in younger to older skin (Adapted from "skin epithelium", by BioRender [29]).

## 2. Mechanisms of Potential Therapeutics and Cosmetic Agents against Aging

The broad interest in the identification and elucidation of effective skin aging treatments is reflected by the amount of reports in the literature. As a result, a wide range of therapeutic agents originating from different sources have been evaluated for their anti-aging properties.

These agents range from plant and algae extracts to cultured neonatal foreskin fibroblast medium [10,40,41]. This part of the review will focus on those agents where clinical evidence or strong in vitro or preclinical data support their use in skin aging, and will discuss the underlying mechanisms if known. Importantly, formulations and delivery methods will be considered, particularly with regard to the site of action and potential effects. Because aging is associated with a decline in function, many anti-aging treatments are directed at replenishing the skin with agents thought to boost or complement intrinsic processes (such as antioxidants), replace those that may be lacking (such as vitamins and minerals), or utilize nature's adaptation to assault (such as polyphenols and carotenoids) [12,31,32]. Detailed summaries are available for retinoids, hyaluronan, astaxanthin, carnosine, lutein, ubiquinone, resveratrol, cosmeceutical peptides, and MMP inhibitors including related anti-aging properties, mechanisms, and/or treatments [7,37,41–51]. The agents presented here were chosen as representatives of some of the most commonly used types of anti-aging therapeutics due to them having compelling evidence from the literature, widespread use, or availability as marketed products.

### 2.1. Synthetic and Plant-Derived Products

#### 2.1.1. Synthetic Products

Doxycycline is a tetracycline antibiotic and an innate MMP inhibitor used in the treatment of periodontal disease [52]. It has been reported to be able to inhibit MMP-2 and MMP-1 expression in vivo [53,54]. The key role of the doxycycline-mediated inhibition of MMPs was discussed in the study by Chiarelli et al. Their study focused on the treatment of hypermobile Ehlers–Danlos syndrome (hEDS), which is associated with ECM organization as it is generally characterized by abnormalities in the ECM. Based on the findings from human studies, it was determined that doxycycline, which is generally used against hEDS, restores the organization of the ECM and reverts the dysfunction of dermal fibroblasts [55].

Diclofenac is a non-steroidal anti-inflammatory drug (NSAID) that likely reduces prostaglandin E2 by way of cyclooxygenase inhibition. Diclofenac has been shown to localize in the epidermis synergistically with hyaluronic acid (HA), but not with other

glycosaminoglycans, pharmaceutical gelling agents, or in buffer [56]. At 3% in a 2.5% HA topical gel, diclofenac has been shown in randomized control trials to effectively reduce and clear actinic keratosis lesions that commonly present on the face and hands from excessive UV exposure [56].

### 2.1.2. Vitamins

Many vitamins are important because of their antioxidant capabilities in human systems. They can decrease ROS in human cells, leading to the production of low-activity molecules. Moreover, they help aging skin cells reduce their oxidative damage in such a manner that the generation of key components of skin cells increases. Further studies have revealed new natural products for anti-aging and additional antioxidant effects of vitamins C, D, A, B12, B3, and E, as well as lipoic acid and coenzyme Q10.

Vitamin D is the most commonly used natural product amongst anti-aging therapeutic agents. Even though UV radiation initiates its biosynthesis, findings show that vitamin D can protect DNA from damage caused by UV radiation, thereby protecting the skin [34]. Vitamin C, or L-ascorbic acid, is an antioxidant that is essential for collagen synthesis, and indirectly scavenges superoxide by formation of an intermediate radical [4,11]. In clinical studies, a 3% oil-in-water (O/W) emulsion was shown to reduce facial wrinkles and a 5% cream was shown to facilitate elastic tissue repair involving dermal papillae [57,58]. Vitamin E, or d-$\alpha$-tocopherol, is an antioxidant that has been shown to inhibit molecularly induced oxidative changes including AP-1 binding in UV-irradiated keratinocytes in a dose-dependent manner [59]. Vitamin E upregulates antioxidant enzymes, scavenges superoxide, and inhibits lipid oxidation [11]. Most research involves mixtures of vitamin E with other antioxidants, such as vitamin C and coenzyme Q10 [8,50].

The regulation mechanism of MMPs is studied during the formulation development of natural compounds, such as folic acid and vitamin B12, which are expected to work as AhR antagonists. According to the study by Kim et al. [37], to impede the formation of wrinkles and DNA damage mediated by MAPK pathways and associated with the induction of UV irradiation, the AhR antagonists should be used via transdermal delivery against skin aging.

### 2.1.3. Endogenous Compounds

Ubiquinone or coenzyme Q10 (CoQ10) has a similar structure to vitamin K, but is synthesized throughout the body, is fat-soluble, and functions as both a pro- and antioxidant [50]. The actual mechanism of CoQ10 against skin aging is to maintain the skin cell organization by impeding the generation of ROS, which is known to regulate the production of MMPs via the activation of the MAPK pathway. Thus, some formulation studies focused on the reduction in MMP-1 expression after UVA exposure in order to demonstrate the efficacy of topical administration of CoQ10. Oral supplementation of CoQ10 when using its water-soluble formulation with improved bioavailability was shown to reduce wrinkles and increase smoothness [60,61].

Retinoids belong to the vitamin A family and are commonly used against skin aging as natural compounds. They are commonly known as retinoic acid, which has the same molecular and functional properties as all related retinol compounds, and prevents the skin from deformation by regulating MMPs [62].

Tretinoin, or all-trans-retinoic acid, is a vitamin A metabolite that can modulate cellular programming in the skin through retinoic acid receptors (RARs) and retinoid X receptors (RXRs) [25], such as by inhibiting the UV-induced activation of NF-κB and AP-1 [45]. The use of retinoids results in epidermal thickening, SC compaction, and synthesis of glycosaminoglycans, thereby decreasing the signs of photoaging in vivo, but with the unfortunate side effects of a burning sensation and dryness [45].

Hyaluronic acid (HA), a major polyanionic glycosaminoglycan found in the ECM, is composed of N-acetyl-d-glucosamine and β-glucuronic acid, and has been shown to uniquely enhance SC penetration and localization of certain therapeutic agents to the epi-

dermis, including clindamycin, cyclosporine, and select NSAIDs [56]. The exact mechanism is unknown, but may involve HA receptors within the skin and the facilitation of drug retention within the HA-hydrated epidermal layers [56,63].

### 2.1.4. Carotenoids

Astaxanthin, a natural pigment synthesized by yeasts, bacteria, plants, and microalgae such as Haematococcus pluvialis, has been shown in a number of studies to significantly improve clinical signs of photoaged skin related to wrinkles, elasticity, and moisture when administered topically and/or orally [46]. With a greater antioxidant capacity than β-carotene, astaxanthin has been shown to block NF-κB activation and inhibit MMP-3 and MMP-1 expression in vitro, leading to reduced inflammation and increased collagen content [64]. Lutein, another xanthophyll carotenoid capable of filtering blue light, has been demonstrated to be protective against skin damage caused by UV radiation, likely through its role as an antioxidant [48].

### 2.1.5. Polyphenols

Resveratrol is an all-natural stilbene that can be extracted from grapes and is thought to reduce signs of aging processes in skin through the inhibition of apoptotic occurrences and mitochondrial dysfunctions [65]. Resveratrol has been shown to modulate inflammatory cytokines such as IL-6, IL-8, and tumor necrosis factor-alpha (TNF-α) in human keratinocytes, thanks to the production of phosphorylated EGFR (epidermal growth factor receptor) [6].

Epigallocatechin gallate (EGCG), a tannin from green tea extract, is an antioxidant that has been shown to reduce oxidative stress and inhibit NF-κB in vitro for protection against UV radiation [6]. Although claimed to provide anti-aging benefits for the skin, minimal human clinical data have been published, including one study that evaluated UV-induced damage for a broad-spectrum sunscreen with and without EGCG. While sunscreen alone decreased the amount of MMP-1 detected in the skin, the addition of EGCG resulted in a significantly greater reduction in MMP-1 [66,67].

Apigenin, a flavone found in many plants, is reported to have anti-inflammatory and antioxidant effects in vitro [68]. When administered to mouse skin inflamed by UVB light, apigenin-containing ethosomes were shown to reduce cyclooxygenase-2, an important enzyme in the synthesis of prostaglandins [69]. The function of apigenin was recently demonstrated in the study by Che-Hwon Park et al., which focused on the mechanism of skin disease and the evaluation of apigenin-mediated amelioration. This study showed that the production of nitric oxide (NO), the expression of cytokines (IL-1, IL-4, IL-5, IL-6, and COX-2), and the phosphorylation of MAPKs were significantly inhibited by apigenin; therefore, it has good potential as a therapeutic agent for autoimmune diseases such as psoriasis [70,71].

Baicalein and baicalin, flavones found in Scutellaria baicalensis, are reported to have a good range for inhibiting the mechanism of not only cancer, but also UVA exposure, ROS, and bacterial infections via LI-promoted Fenton chemistry [71]. The extract of S. baicalensis was shown to diminish the excessive release of oxidative stress and to neutralize the production of ROS. Thus, the main mechanism of baicalin has been related to its anti-inflammatory function owing to the inhibition of NF-κB, COX-1, and iNOS [71,72].

## *2.2. Peptides, Cell-Derived Products, and Biologics*

### 2.2.1. Peptides, Proteins, and Cell Culture-Derived Extracts

Carnosine, an endogenous dipeptide (β-alanyl-l-histidine) that is synthesized in muscle and brain cells, has been credited with cell-regenerating and lifespan-extending properties in cultured human fibroblasts, but the evidence beyond preclinical studies is limited [47]. Data suggest that carnosine administration can suppress the growth-inhibitory cytokine TGF-β, which inhibits telomerase, as well as increase circulating IGF-1, which is associated with reduced wrinkles [73,74].

Recently, Aldag, Teixeira, and Leventhal summarized clinical evidence for selected peptides and proteins used cosmetically for aging, including matrikines, matrikine-like peptides, growth factors, cytokines, and protein extracts, complementing the previous summary by Gorouhi and Maibach [40,42]. Clinical investigations have been performed to evaluate the efficacy of some of these therapeutic agents (Table 1), which supports the label claims on related marketed products.

As shown in Table 1, many of these products contain matrikines or matrikine-like peptides. As matrikines are peptide fragments released through the proteolysis of ECM components that can trigger the synthesis activity of collagens, elastin, and glycosaminoglycans, they have proven useful in anti-aging treatments [75]. These peptides have beneficial growth factor-like activities, but are much smaller and simpler to utilize for formulation purposes [76]. Minor modifications to the peptide structures, such as by adding palmitoyl to KTTKS or octanoyl to carnosine, have led to significant improvements in skin permeation, and thus, to efficacy [75,77].

Table 1. Products containing matrikines and matrikine-like peptides.

| Brand | Ingredient(s) | Origin | Cosm. Use | Mechanism | Delivery System | Ref. |
|---|---|---|---|---|---|---|
| Iamin | GHK-Cu or copper tripeptide-1 | fragment of the α2-chain of collagen I | anti-aging | enhances $Cu^{2+}$ uptake, TIMP-1, TIMP-2, and MMP-2; promotes degradation of collagen aggregates; promotes collagen, elastin, proteoglycan, and glycosaminoglycan production; anti-inflammatory and antioxidant | hydrogel alternative: microneedles | [42,78–80] |
| Matrixyl | pal-KTTKS, palmitoyl pentapeptide-4, palmitoyl pentapeptide-3, or palmitoyl oligopeptide | procollagen I-derived pentamer (KTTKS) with palmitoyl added to improve permeability | anti-aging | promotes the production of fibronectin, elastin, glycosaminoglycans, and collagen types I, III and VI; stabilizes mRNAs that upregulate TGF-β | O/W emulsion moisturizer | [42,77,81] |
| Preregen | soybean protein, glycine soy protein, and amino acids | soybean seed | anti-aging | inhibits proteinase formation and increases number of dermal papillae | O/W emulsion in Tegocare-45 base | [42,82] |
| Keramino 25 | keratin protein and amino acids | human hair and sheep wool | anti-aging moisturizer | improves elasticity and hydration of the skin and hair | multilamellar vesicle liposomes, 0.9% NaCl | [42,83] |
| Citrix CRS | cell rejuvenation system (L-ascorbic acid, TGF-β1, and Cimicifuga) | recombinant TGF-β1, stem cell extract from Cimicifuga racemosa | anti-aging | profibrotic cytokine modulates angiogenesis, cellular migration and proliferation, neocollagenesis, and degradation of matrix proteins | liposome cream in silicone base | [42,84] |
| Processed skin cell proteins (PSP) | PSP bio-restorative skin cream (mixture of cytokines and growth factors) | human fibroblast cell culture lysate | anti-aging | growth factors promote angiogenesis and cytokines modulate inflammation | cream | [14,40,85] |
| Nouricel-MD TNS | tissue nutrient solution recovery complex (VEGF, PDGF-A, G-CSF, HGF, IL-6, IL-8, and TGF-β1) | human neonatal foreskin fibroblast culture | anti-aging | growth factors promote angiogenesis, and cytokines modulate inflammation and enhance ECM component deposition | oil-free gel | [14,40,42] |
| ReGenica, MRCx | MRCx (VEGF, IL-8, and keratinocyte growth factor, but no TGF-β) | Cytokines and growth factors from conditioned medium of fibroblasts | anti-aging moisturizer | growth factors promote angiogenesis and cytokines modulate inflammation | cream | [40,86] |
| Micro-protein complex (MPC) | GEGK, pal-GHK, and N-octanoyl-carnosine | synthetic | anti-wrinkle | increases ECM collagen, hyaluronan, and fibronectin | cream | [40,76,87] |
| Decorinyl | tripeptide-10 citrulline, or Lys-α-Asp-Ile-citrulline | synthetic decorin-like tetrapeptide | anti-aging | regulates collagen fibrillogenesis, and improves uniformity of and influences diameter and position of collagen fibers | liposomal cream | [42,88] |

**Table 1.** *Cont.*

| Brand | Ingredient(s) | Origin | Cosm. Use | Mechanism | Delivery System | Ref. |
|---|---|---|---|---|---|---|
| Progeline | trifluoroacetyl-tripeptide-2 or trifluoroacetyl-Val-Tyr-Val | synthetic | anti-wrinkle | inhibits MMP and decreases the synthesis of progerin, and improves the relation of collagens via the production of proteoglycan | cream | [89] |
| SYN®-AKE | tripeptide-3 or dipeptide diamino butyroyl benzylamide diacetate | Synthetic tripeptide mimicking Waglerin-1 from snake venom | anti-wrinkle | formulated for the treatment of neuromuscular activity; inhibits the muscular activity associated with repeated movement at the neuromuscular junction | Hydrogel = glycerin-based aqueous solution | [89–91] |

### 2.2.2. Biologics and DNA Repair

Botulinum toxin (BTX) is generated in the anaerobic spore of the Clostridium bacteria. This toxin is a complex mixture of botulinum neurotoxin and several non-toxic proteins [92]. Botulinum toxin has eight distinct antigenic profiles (A–G), but their presence depends on the different strains of Clostridium botulinum. The human nervous system can develop botulism if it comes in contact with the A, B, C, E, F, and G strains, but it is unaffected by the D strain [93]. BTX is synthesized as a single-chain polypeptide that becomes a double chain with a disulfide bridge thanks to the action of proteases (a process known as activation) [92,93].

SP1, SP2, and SP3, which are high-affinity monoclonal antibodies targeting mouse MMP-1A, MMP-2, and MMP-3, respectively, were made using protein engineering from scFv fragments identified using phage-display library screening experiments [94]. There was no cross-reactivity, low nanomolar binding affinity (KD 6 nM), or low or undetectable expression of the MMP-1A antigen in healthy tissue, indicating the excellent potential of the SP1 antibody in disease-targeting applications, which could translate to advancements in skin aging and possibly new effective and non-toxic MMP inhibitor treatments [38].

T4 endonuclease V, or T4N5, is a bacteriophage-derived DNA repair enzyme that has been shown to reduce cyclobutane pyrimidine dimers (CPDs) and MMP-1 activation caused by UV irradiation in keratinocytes [4]. When administered topically after UV exposure in a randomized clinical study as pH-sensitive liposomes in a hydrogel lotion, the removal of UV-induced CPDs was accelerated compared to the placebo [95].

## 3. Formulation Approaches and Delivery Strategies—Patents, Papers, and Products

Complementary to the identification of effective therapeutic agents for aging-related treatments, there is a need for robust formulations and suitable delivery systems to employ those agents. As such, there are detailed reviews on delivery systems for cosmetic or cutaneous use [96–100], preferred materials used in these delivery systems [56,101], and specific classes of anti-aging compounds, such as polyphenols or antioxidants [11,17,102,103].

### 3.1. Common Formulation Approaches for Topical Administration

While data support the fact that oral supplementation is a useful route to deliver vitamins, minerals, phytochemicals, and carotenoids to the skin, cosmetic products mainly rely on the topical route of administration [17,104]. This is a logical approach given that the sites of action for most skin aging-related issues reside within the epidermal or dermal layers, and therefore, those formulation strategies amenable to topical delivery are considered here. Delivery systems described in the literature for topical administration include emulsions, liposomes, niosomes, ethosomes, transfersomes, solid lipid nanoparticles (SLNs), nanostructured lipid carriers (NLCs), microparticles, nanoparticles, and microneedles [17,97–99,105,106].

### 3.1.1. Emulsions

Emulsion systems are composed of two continuous phases: oil-in-water (O/W) or water-in-oil (W/O). Microemulsions (size range from 10 to 100 nm) differ from traditional emulsions and they are thermodynamically stable mixtures of oil, water, and surfactant that can be prepared from simple mixing using ionic or non-ionic surfactants [107]. Therefore, they can be ternary (with surfactant) or quaternary (with a co-surfactant if necessary). Generally, microemulsions applied via topical administration are in the O/W phase, which permits them to achieve penetration of a lipophilic drug through the stratum corneum [108]. There are several exceptions according to composition, temperature sensitivity, and larger surfactant content that can lead to shelf-instability and skin irritation issues [98,109]. To enhance the stability of the formulation and impede the possibility of coalescence, a suitable surfactant should be chosen carefully. The proper surfactant creates the necessary balance among water and oil substances. In contrast, nanoemulsions are kinetically stable mixtures prepared via higher energy methods at lower surfactant levels, with typical droplet sizes

ranging from 100 to 400 nm [98,107,109]. However, some disagreements have evolved recently regarding their thermodynamic stability as compared to that of microemulsions. Due to the small droplet size of the nanoemulsions, gravity has a minimal effect on the particles, and Brownian motion tends to overrule the kinetic stability leading to coalescence [108,110]. In addition to the increase in kinetic stability, the interaction among the small droplets contributes to the balance between phases against sedimentation, which is unlike the thermodynamic stability of microemulsions where temperature is a limiting factor [111]. Thus, more stable formulations can be achieved, for example, by incorporating the emulsion in a gel matrix, forming an emulgel or nanogel [112,113].

The key part of the gel formation is the addition of a gelling agent which aids in improving the stability and bioavailability of the system. This formulation includes polymerization, i.e., the cross-linked or self-assembly interaction, of polymers into a micro- or nanogel [113]. The most frequent polymers used in topical administration are natural biodegradable polymers such as hyaluronic acid, dextran, chitosan, alginate, and poly-γ-glutamic acid. Furthermore, there are several synthetic polymers which are commonly used due to their biocompatibility such as polylactic acid, polyglycolic acid, and polylactic-co-glycolic acid [114].

### 3.1.2. Vesicular Systems

Liposomes have an aqueous core surrounded by one or more lipid bilayers formed from phospholipids and can accommodate both hydrophilic molecules (in their aqueous core) and lipophilic compounds (within the bilayer) (Figure 3a) [115]. Shortcomings of this delivery system include instability in biological fluids and burst release upon storage, as well as expensive manufacturing due to low reproducibility [103,115]. Niosomes (Figure 3a) are also vesicles that can accommodate hydrophilic and hydrophobic compounds, and have an aqueous core surrounded by non-ionic surfactants (usually Span®, Tween®, or Brij®) which act as skin penetration enhancers and increase skin permeability [116]. They typically include cholesterol or a derivative that increases the phase transition temperature (TC) of the lipids and can improve stability, entrapment efficiency, and drug release properties of actives from the vesicles [116].

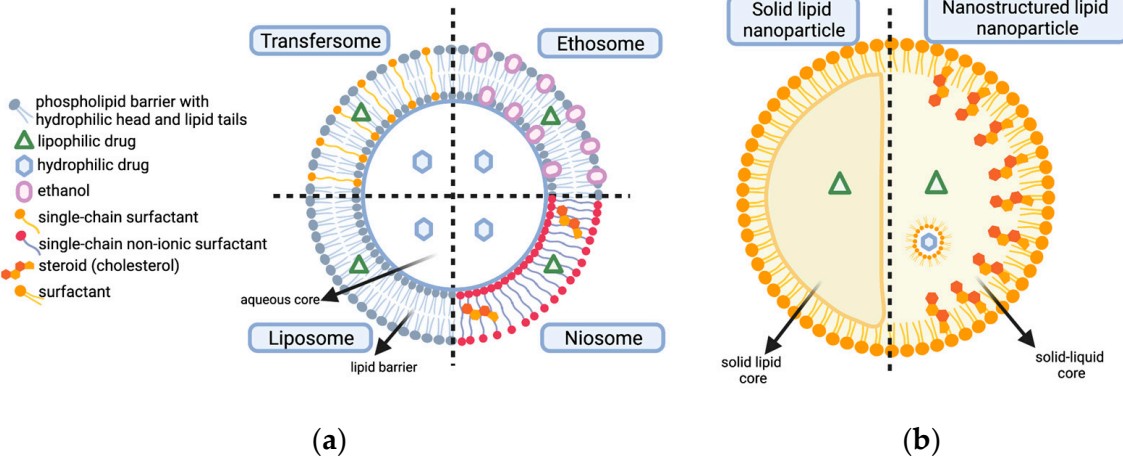

**(a)**         **(b)**

**Figure 3.** Vesicular (**a**) and particulate (**b**) systems (Illustration created with BioRender [29]).

Ethosomes (Figure 3a) are flexible vesicles formed from the combination of phospholipids, ethanol, and water, and can solubilize poorly soluble actives [99]. Ethanol, a known penetration enhancer, exhibits synergistic effects with the vesicles and/or skin lipids for improved delivery into the skin [117]. Transfersomes (Figure 3a) are highly deformable vesicles made of phospholipids, surfactants, and water, and are capable of penetrating through the SC into the skin [118] The surfactant, referred to as the edge activator, usually has a single chain and acts as a penetration enhancer by decreasing the bilayer rigidity and enabling the vesicles to remain intact while traversing the SC [99].

### 3.1.3. Particulate Systems

Solid lipid nanoparticles (SLNs) (Figure 3b) combine the advantages of emulsions' and liposomes' biocompatible excipients and the ease of manufacturing with that of solid particles' protective encapsulation and drug release modulation [96]. Having a mean particle size of ~40–1000 nm, they likely facilitate occlusive properties and deeper penetration into the skin, but shelf-stability is an issue due to crystallization of the solid lipid core [17,96,97]. Nanostructured lipid carriers (NLCs) (Figure 3b) overcome the low drug loading and shelf-stability issues of SLNs by using liquid and solid lipid blends that are solid at body temperature [96]. In addition to the benefits of SLNs, they contain less water and can be designed to have more tailored release profiles depending on the particle size, exact lipid composition, and encapsulated drug [99].

Polymeric microparticles and nanoparticles made from biodegradable polymers enhance material handling properties and the stability of the encapsulated active ingredient, as they tend to accumulate in hair follicle canals [97,101]. Microparticles can be prepared from solutions, suspensions, and emulsions by spray-drying, which is one of the most commonly employed manufacturing techniques [17].

### 3.1.4. Microneedles

Microneedles are meant to go through the stratum corneum of the skin and create micron-sized pathways through the epidermis or upper dermis, from where the formulated product can either be delivered locally or injected directly into the blood flow without any obstacles. Microneedles combine the advantages of transdermal and intravenous administration, and are utilized in cosmetic applications before surmounting the clinical hurdle for use in vaccine delivery (with or without adjuvants) [105]. Disadvantages vary by type, but include low drug loading, as it is limited by the total encapsulated or coated amount, non-continuous batch manufacturing processes, and the lack of safety data for the potential detachment of the needles from the device/array within the skin [106].

Microneedle patches comprise a series of arranged micron-sized needles (Figure 4) in square patches, which are similar to transdermal patches that contain the drug. These needles go through the stratum corneum, allowing for the release of the drug without stimulating the pain receptors [119]. The main mechanism of drug delivery is either by active or passive diffusion after disrupting the skin and the use of a permeation enhancer [120]. This technique can be useful for the delivery of mainly hydrophilic molecules through the skin. These needles can be coated with the drug, be hollow (and the drug is stored in a reservoir) or dissolving, or they may form a hydrogel that is interlocked with the subdermal layer of the skin, releasing the drug over time [106,121] (Figure 4).

### 3.2. Common Formulation Delivery Methods Used against Skin Aging

Topical therapeutics and cosmetic agents are normally presented as creams, gels, sprays, lotions, and parenteral preparations. They consist of a suitable base where one or more active ingredients can be dissolved or uniformly dispersed together with any suitable excipients such as emulsifiers, viscosity-increasing agents, antimicrobial agents, antioxidants, or stabilizing agents.

### 3.2.1. Creams, Gels, and Serums

Creams are biphasic and semi-solid products. There are several studies on the efficacy of cream preparations on human skin in order to understand how they increase skin protection or impede the aging process. An open study by Goffin et al. [122] compared the efficacy of vitamin E and 0.075% retinol with patients who volunteered to follow a protocol of a cream containing vitamin E and retinol. At the end of the study, superficial wrinkles caused by UV radiation showed that retinol was more efficient than vitamin E [122]. The development of a formulation studied by Kaci et al. [123] showed that CoQ10 can be formulated using nanoemulsions in the cream preparation. Their in vitro biocompatibility test concluded that the antioxidant effect of CoQ10 aided in protecting DNA; in addition,

xanthan gum, which was used as a texturing agent, enhanced the glucose metabolism by leading to improved cell growth. Moreover, the nanoemulsion cream formulation favored the dispersibility of CoQ10 by confirming the cell proliferation [123]. Besides the common emulsion preparations of creams, there are other studies on the formulation development of dispersions made with SLNs or NLCs by directly mixing with oil-in-water or by replacing the aqueous part of an oily cream [124]. Although these studies seem more complicated due to the stability problems of emulsion systems and considering their cosmetic purposes, great attention has been given to the synergistic moisture effect [100]. Therefore, widespread preparations of nano-based anti-aging creams have become more popular, and there are several products that have been commercialized. The most well-known NLCs on the market were launched by Dr. Rimpler GmbH, Lancôme, and Yamanouchi [114,125].

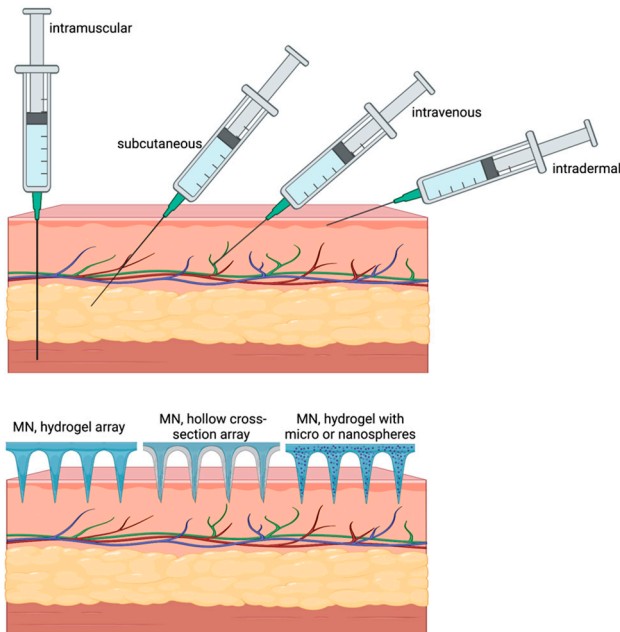

**Figure 4.** Parenteral and microneedle injections (Illustration created with BioRender [29]).

Gels are commonly used semi-solid products for skin protection and anti-aging. Since the idea of nano-based preparations against aging seems promising, some scientists have developed semi-solid formulations by replacing the lipid part of nanoparticle dispersions with hydrophilic gelling agents such as carbomer, methyl cellulose (MC), hydroxypropyl cellulose (HPC), and hydroxypropyl methyl cellulose (HPMC) via their direct addition to the formulation [100,126]. For instance, the evaluation of an SLN dispersion and hydrogel with vitamin A palmitate performed by Jeon et al. demonstrated the anti-wrinkle effect of this compound [127]. One study compared the gel and cream formulations prepared using SLNs [114]. The addition of different SLN formulations to a conventional cream improved skin hydration, while the SLN-based tretinoin gel improved the topical administration of tretinoin compared to the conventional formulations on the market [114]. Another study on the combination of a hydrogel and encapsulated niosomes with curcumin, which is a natural polyphenol found in Curcuma longa roots, showed an anti-wrinkle effect as well as anti-cancer and anti-inflammatory properties [128]. This type of gel preparation is commonly selected due to the convenient application of the gel by the patients. Thus, the cosmeceutical companies developed the gel products with nano-based dispersions with different concentrations of gelling agents inside the nanodispersions.

A serum is an oil (and less commonly, water)-based formulation that contains peptides or oil meant to be absorbed by the skin. Other components may be retinol, hyaluronic acid, or different kinds of lipophilic vitamins. The common serums and gels against aging that

are commercialized include the NanoRepair Q10 serum by Dr. Rimpler GmbH, HydraZen serum by L'oreal, and Hydro Boost water gel by Neutrogena [128].

### 3.2.2. Sprays and Lotions

Sprays and lotions are among the most common cosmetic formulations used for skin protection and anti-aging products which are manufactured using nanoemulsions [18]. For instance, the study by Piccioni et al. aimed to evaluate their photoaging therapy with liposome-encapsulated sprays loaded with 5-aminolevulinic acid using an intense pulsed light technique [129]. Their study was conducted with healthy volunteer patients, aged from 35 to 65, who visited the clinic every 3 weeks, with a final control 3 months after the end of the treatment. The improvement in photoaging and its associated side effects, namely, wrinkle reductions, were observed using liposome-encapsulated 5-aminolevulinic acid [129]. On the other hand, lotions are described as an anti-aging product, even though they are used in moisturizers or sunscreens. Their formulations consist of nanoparticles; however, they have a high water content and less viscoelastic behavior compared to the other semi-solid products. A study by Han et al. compared the anti-aging effect of the liquid protein solutions and the lotion formulated with three peptides (acetyl hexapeptide-3, carnosine, and palmitoyl tripeptide-5) [130]. They highlighted the easy application and spreadability of their new lotion formulation with an anti-aging reduction, and found that wrinkles decreased by 30.8% [130]. The most common spray and lotion formulations are sunscreen products and, given that they protect the skin from the aging effect of UV rays, they can be considered as anti-aging products.

### 3.2.3. Parenteral Preparations

Parenteral preparations for therapeutic and cosmetic agents are traditionally considered the most effective route of drug administration as they prevent the possible loss of efficacy due to first-pass metabolism or the proteolytic cleavage of peptides and proteins [91]. However, common problems associated with hypodermic needles include pain and anxiety. Their administration can be divided into four most used routes, including intradermal, intravenous, intramuscular, and subcutaneous injections [121]. The needle length mostly depends on the administration route, as the deeper injection type is related to a longer length and shorter gauge. Thus, recent advancements have been gravitating towards microneedle injector applications which penetrate the epidermis via intradermal injection. The formulations on the market are mostly meant for cosmetic use and contain hyaluronic acid, but other purposes are under investigation (vaccines). BTX is used in various fields of medicine, including the treatment of hyperhidrosis and cervical dystonia. The recommended injection route is usually intramuscular or intradermal [92,93]. Botox®, Dysport®, Xeomin®, and NeuroBloc® are commercially available and each one of them have their own formulation and dosage. Local injections of BTX prevent the release of acetylcholine and co-transmitters in peripheral cholinergic nerve endings, which consequently leads to reduced symptoms [93,131].

BOTOX® Cosmetic for injection, a sterile, vacuum-dried purified botulinum toxin type A, is produced from the fermentation of the Hall strain of Clostridium botulinum type A grown in a medium containing casein hydrolysate, glucose, and yeast extract, and is intended for intramuscular use [92,131].

### 3.3. Limitations on Current Approaches/Agents

#### 3.3.1. Permeation Enhancers

Cosmetically relevant permeation enhancers include alcohols, glycols, fatty acids, phospholipids, Azone, urea and derivatives, cyclodextrins, and dimethylsulfoxide [99,101,132]. Surfactants also act as permeation enhancers, both of which are commonly employed in topical delivery systems often to overcome the physicochemical limitations of the active or SC barrier properties, but must be chosen carefully due to their potential to cause skin irritation [63,133].

Non-ionic surfactants, such as those used in niosomes, with a polar head group and long alkyl chain have been shown to disrupt lipids of the SC [63]. Other permeation enhancers, such as oleic acid, also disrupt the SC lipids, but appear to be deposit rather than dispersing forms [63]. Products containing high amounts of ethanol, such as ethosomes, may enhance therapeutic delivery through the skin by altering the lipid content of the SC, but they could also increase the solubility of the drug within the SC [63,134]. SC alterations, even if meant to be transient, are more likely to be associated with increased irritation [133].

### 3.3.2. Systemic Delivery

If not chosen carefully, permeation enhancers could lead to systemic delivery, which is generally avoided for cosmetic and topical treatments [17]. Specifically, due to the incorporation of potent edge activators or ethanol, transfersomes and ethosomes can enable transdermal delivery of actives to the systemic circulation and overshoot epidermal sites of action, which could lead to undesirable side effects [69,118]. Similarly, microneedles can also lead to systemic drug delivery, and thus should be designed carefully to retain the drug in the superficial layers of the skin, possibly by using shorter microneedles [135]. For delivery systems that are designed to target the SC, such as liposomes, bioavailability in the skin is difficult to determine. Tape-stripping is a minimally invasive technique for SC sampling that is used to evaluate topical drug exposure [133].

### 4. Conclusions

Recently, a bounty of information regarding anti-aging and cutaneously active therapeutics has been made available in the literature, along with applicable delivery systems to target the relevant sites of action. While MMP inhibitors hold immense potential for anti-aging treatments, the myriad complex ways in which MMPs function are still not well-understood. Additionally, though Levin et al. [136] has suggested a "short term topical treatment" as the ideal administration regimen, related research has not focused on the most prominent MMP inhibitors for skin aging.

While promising, clinical data for the more recently designed, highly selective MMP inhibitors are limited. Similarly, for some notable conventional, anecdotally trusted anti-aging compounds, such as EGCG, clinical data are either uncompelling or lacking. By applying enabling technologies such as the delivery systems outlined here, existing agents can be repurposed or fine-tuned, and traditional but unproven treatments can be optimized for efficacious dosing and safety.

**Author Contributions:** Conceptualization, A.A.B. and T.T.; methodology, A.A.B. and T.T.; investigation, A.A.B., T.T. and C.B; writing—original draft preparation, A.A.B.; writing—review and editing, A.A.B. and C.B.; supervision, Giorgia Pastorin and Hugh DC Smyth. All authors have read and agreed to the published version of the manuscript.

**Funding:** This work was supported by the Industry Alignment Fund—Pre-Positioning (IAF-PP) grant (A20G1a0046 and A-0004345-00-00), Singapore.

**Institutional Review Board Statement:** Not applicable.

**Informed Consent Statement:** Not applicable.

**Data Availability Statement:** Not applicable.

**Acknowledgments:** The authors would like to thank Hugh DC Smyth (College of Pharmacy, University of Texas at Austin) and Giorgia Pastorin (Department of Pharmacy, National University of Singapore) for kindly suggesting some of the topics covered in this review. This work was supported by the National University of Singapore (NanoNash Program A-0004336-00-00, & MOE Tier 1 A-0008504-00-00). Giorgia Pastorin would also like to thank the Industry Alignment Fund—Pre-Positioning (IAF-PP) grant (A20G1a0046 and R-148-000-307-305/A-0004345-00-00). A.A.B. would like to thank the IAF-PP grant grant (A20G1a0046 and R-148-000-307-305/A-0004345-00-00) for the support for her research position at the National University of Singapore.

**Conflicts of Interest:** The authors declare no conflict of interest.

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
