# Peer review of "Current Insights into the Formulation and Delivery of Therapeutic and Cosmeceutical Agents for Aging Skin"

_cosmetics, doi:10.3390/cosmetics10020054_

Round 1
Reviewer 1 Report
In this manuscript, the authors tried to summarize the formulation and delivery of therapeutic and cosmeceutical agents for aging skin. The topic is very meaningful and the structure of this review is very well organized. Therefore, this review may be a good contribution to the field of cosmetics and a valuable asset for the reader. Some specific suggestions follow:
1. Although the 4 figures were created with [29] Biorender, Biorender may be not suitably listed as a reference. It is recommended to indicate them in the figure caption instead.
2. In “Section 2.1.1. Small molecules”, only one compound “doxycycline” is introduced. Are there more examples? Additionally, small molecules usually include vitamins. So, are vitamins suitable for listing outside of 2.1.1? Maybe “Synthetic products” and “Synthetic and plant-derived products” are more suitable for 2.1.1 and 2.1, respectively.
Author Response
- Although the 4 figures were created with [29] Biorender, Biorender may be not suitably listed as a reference. It is recommended to indicate them in the figure caption instead. R: Thank you so much for the suggestion. 29 is the number of the cited online application that we used for the images. We are paying the license to create our own images under our name.
2. In “Section 2.1.1. Small molecules”, only one compound “doxycycline” is introduced. Are there more examples? Additionally, small molecules usually include vitamins. So, are vitamins suitable for listing outside of 2.1.1? Maybe “Synthetic products” and “Synthetic and plant-derived products” are more suitable for 2.1.1 and 2.1, respectively. R: Thank you so much for the suggestion. Doxycycline is not the only one, diclofenac is also one of them.
I would like to thank for your directions. All corrections are conducted. The revised version is in the attached file.
Reviewer 2 Report
This manuskript does not add to the body of literature in the field of cosmetology or dermatology. Moreover, it is written in poor English language and has major grammatical mistakes, or missing punctuation.
Also, I believe it is not acceptable for publication in this form as most of the figures are from single article (29).
Author Response
This manuskript does not add to the body of literature in the field of cosmetology or dermatology. (R: We are not aiming that. We would like to describe the current state of the art in SC delivery, with particular interest in nanoscale, where we discuss the most significant advances with the aim of stimulating the search for new and more effective formulations with natural products and peptides.)
Moreover, it is written in poor English language and has major grammatical mistakes, or missing punctuation. (R: We do not agree with this point. Although one of the authors (Tamara Tarbox) is a US citizen and native English speaker, we used the english editing service. We would like to know if you have any specific grievance on the wording used for the article. )
Also, I believe it is not acceptable for publication in this form as most of the figures are from single article (29). (R: 29 is the number of the cited online application that we used for the images, so it is not an article. We are paying the license to create our own images under our name. The online application name is Biorender.)
Round 2
Reviewer 2 Report
I believe manuscript is now suitable for publication